# Materials and Manufacturing Techniques for Polymeric and Ceramic Scaffolds Used in Implant Dentistry

**Mutlu Özcan** [1,*], **Dachamir Hotza** [2], **Márcio Celso Fredel** [3], **Ariadne Cruz** [4]
**and Claudia Angela Maziero Volpato** [4]

1   Center of Dental Medicine, Division of Dental Biomaterials, Clinic for Reconstructive Dentistry, University of Zürich, Plattenstrasse 11, CH-8032 Zürich, Switzerland
2   Technological Center, Department of Chemical Engineering, Federal University of Santa Catarina, 88040-900 Florianópolis, Santa Catarina, Brazil; dhotza@gmail.com
3   Technological Center, Department of Mechanical Engineering, Federal University of Santa Catarina, 88040-900 Florianópolis, Santa Catarina, Brazil; m.fredel@ufsc.br
4   Health Sciences Center, Department of Dentistry, Federal University of Santa Catarina, 88040-900 Florianópolis, Santa Catarina, Brazil; ariadne.cruz@ufsc.br (A.C.); claudia.m.volpato@outlook.com (C.A.M.V.)
*   Correspondence: mutluozcan@hotmail.com; Tel.: +41-44-634-5600

**Abstract:** Preventive and regenerative techniques have been suggested to minimize the aesthetic and functional effects caused by intraoral bone defects, enabling the installation of dental implants. Among them, porous three-dimensional structures (scaffolds) composed mainly of bioabsorbable ceramics, such as hydroxyapatite (HAp) and β-tricalcium phosphate (β-TCP) stand out for reducing the use of autogenous, homogeneous, and xenogenous bone grafts and their unwanted effects. In order to stimulate bone formation, biodegradable polymers such as cellulose, collagen, glycosaminoglycans, polylactic acid (PLA), polyvinyl alcohol (PVA), poly-ε-caprolactone (PCL), polyglycolic acid (PGA), polyhydroxylbutyrate (PHB), polypropylenofumarate (PPF), polylactic-co-glycolic acid (PLGA), and poly L-co-D, L lactic acid (PLDLA) have also been studied. More recently, hybrid scaffolds can combine the tunable macro/microporosity and osteoinductive properties of ceramic materials with the chemical/physical properties of biodegradable polymers. Various methods are suggested for the manufacture of scaffolds with adequate porosity, such as conventional and additive manufacturing techniques and, more recently, 3D and 4D printing. The purpose of this manuscript is to review features concerning biomaterials, scaffolds macro and microstructure, fabrication techniques, as well as the potential interaction of the scaffolds with the human body.

**Keywords:** biomaterials; bone grafts; bone repair; dental implants; scaffolds

## 1. Introduction

Osseointegration and dental implants were introduced in dentistry more than 40 years ago, thanks to the pioneering studies of Per-Ingvar Brånemark and collaborators [1–3]. Since then, unitary, partial, and total dental losses have been rehabilitated by implant-supported prosthesis successfully and predictably [4]. Dental implants are devices usually made of pure grade IV titanium and are surgically installed in healthy bone areas. After the osseointegration period, which is about 3 to 6 months, they can be restored by the dental prostheses, thus collaborating with the restoration of the masticatory function and the return of oral comfort and aesthetics to the patient [5].

However, the loss of alveolar bone that occurs before or after a tooth extraction is responsible for altering the original volume of the alveolar ridge, as well as for the formation of bone defects (Figures 1–3). After tooth extraction, an average alveolar bone loss of about 30% (in the vertical direction) and 40–50% (in the horizontal direction) occurs for up to 6 months [6]. If no treatment is made, bone loss advances, reaching 40–60% reductions in bone crest volume within 3 years [7].

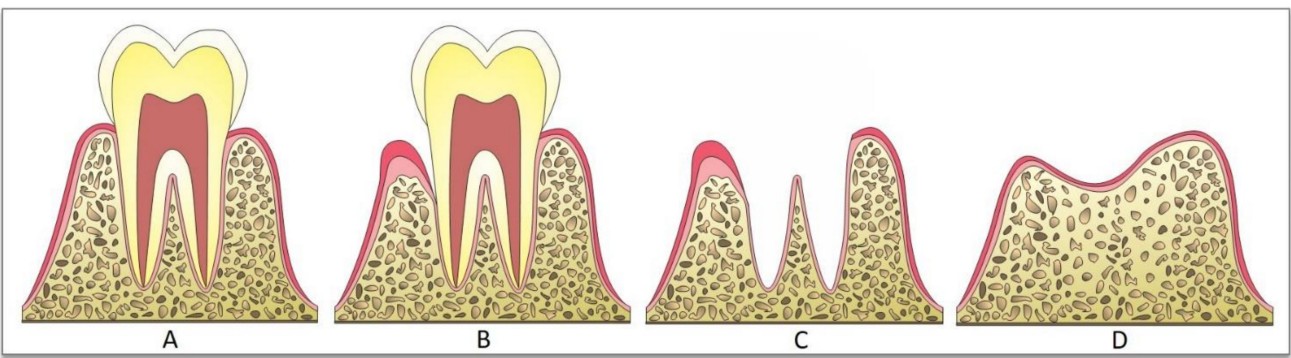

**Figure 1.** Diagram showing the change in the volume of the alveolar bone ridge due to bone loss before tooth extraction. (**A**) Proper dental implantation. (**B**) Alveolar bone loss due to periodontal disease. (**C**) Bone condition after tooth extraction. (**D**) Alveolar bone healing.

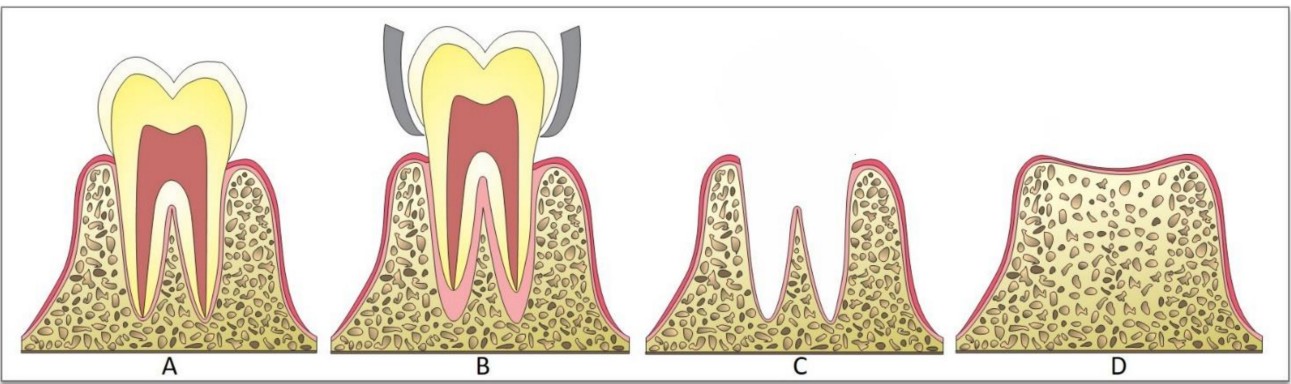

**Figure 2.** Diagram showing the change in the volume of the alveolar bone ridge resulting from the atraumatic tooth extraction. (**A**) Proper dental implantation. (**B**) Dental extraction with no fracture of the alveolar bone wall. (**C**) Bone condition after tooth extraction. (**D**) Alveolar bone healing.

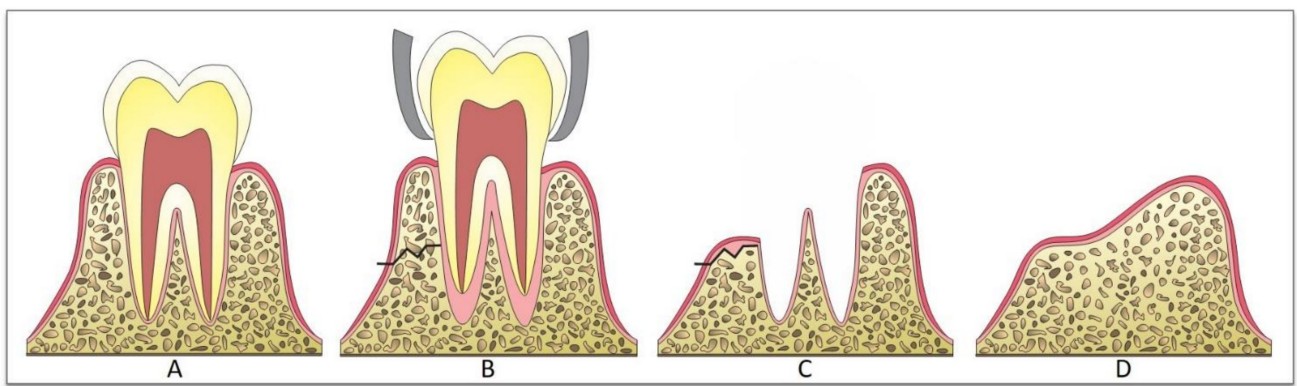

**Figure 3.** Diagram showing the change in the volume of the alveolar bone ridge resulting from the traumatic tooth extraction. (**A**) Proper dental implantation. (**B**) Dental extraction with fracture of the alveolar bone wall. (**C**) Bone condition after tooth extraction. (**D**) Alveolar bone healing.

Bone loss before tooth extraction may be related to periodontal diseases, periapical pathologies, and trauma to the dentition and/or bone [6]. It is important to mention that bone loss after a tooth extraction is also related to the type of the surgical procedure, being aggravated mainly by invasive and traumatic surgeries [8]. In addition, bone loss resulting from tumors or genetic disorders has also been reported [9,10].

Bone remodeling that occurs after tooth loss certainly results in the formation of a bone defect that makes dental implant placement difficult or even unfeasible depending on the size and location [8]. In cases of implants, positioned in deficient bone or extraction

cavities with compromised bone walls, horizontal and/or vertical defects can be formed, exposing the implant body and compromising the short- and long-term functional and aesthetic results [11]. Therefore, the reestablishment and maintenance of the dimensions of the alveolar ridge after tooth loss are essential to ensure a favorable and predictable result with osseointegrated implants [8,12].

Additionally, in order to minimize the aesthetic and functional effects caused by intraoral bone defects, the clinical use of scaffolds as preventive and regenerative techniques have spread widely, enabling the installation of dental implants and, consequently, implant-supported prosthesis rehabilitation. This review describes the features of biomaterials used as scaffolds to promote bone formation, macro and microstructures of scaffolds, fabrication techniques, as well as the potential interaction of the scaffolds with the human body. Scaffolds composed of mainly bioabsorbable ceramics, such as hydroxyapatite (HAp) and β-tricalcium phosphate (β-TCP), and biodegradable polymers like cellulose, collagen, glycosaminoglycans, polylactic acid (PLA), polyvinyl alcohol (PVA), poly-ε-caprolactone (PCL), polyglycolic acid (PGA), polyhydroxylbutyrate (PHB), polypropylenofumarate (PPF), polylactic-co-glycolic acid (PLGA), and poly L-co-D, L lactic acid (PLDLA) are described. This present review also examines hybrid scaffolds that can combine the tunable macro/microporosity and osteoinductive properties of ceramic materials with the chemical/physical properties of biodegradable polymers. Various methods for the manufacture of scaffolds with adequate porosity, such as conventional and additive manufacturing techniques and, more recently, 3D and 4D printing are discussed. Finally, this review briefly discusses the new trends and future directions in developing scaffolds for bone formation and presents relevant information regarding the main materials and manufacturing techniques for scaffolds used in implant dentistry, including the trends in material composition and manufacturing techniques.

## 2. Bone-Grafting Techniques

Preventive (such as atraumatic tooth extraction and filling the socket soon after extraction) or regenerative techniques (such as grafting to gain bone volume after healing the ridge) have been suggested to minimize the esthetic and phonetics effects caused by the bone defects and enable the placement of dental implants [11,13] (Figures 4 and 5). Both techniques employ bone grafts to promote bone repair and the reduction of bone defects. In preventive techniques, the bone grafts help to maintain the volume for cell infiltration and proliferation, as well as assist in closing the surgical wound [14]. In regenerative techniques, the bone grafts have been used to increase the vertical and/or horizontal volume of the alveolar ridge, being the guided bone regeneration (GBR) indicated as the best technique, with satisfactory results over time [15–17].

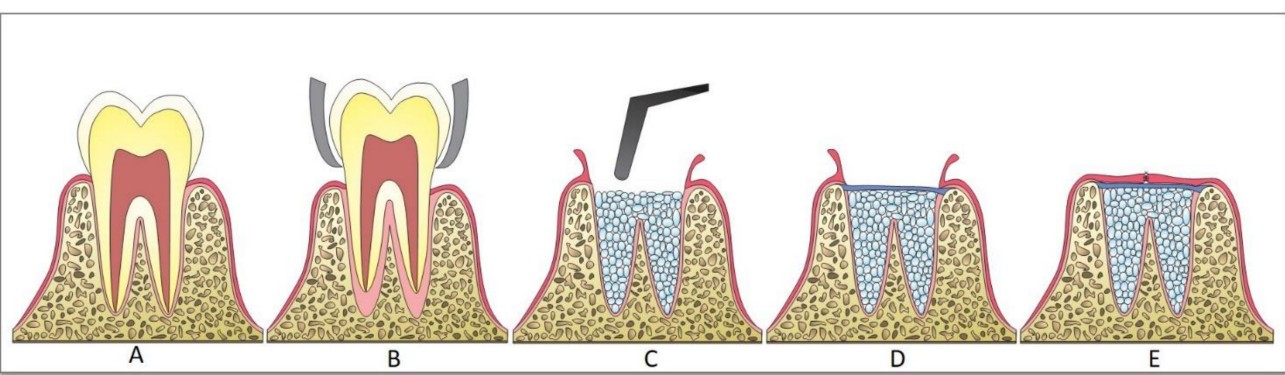

**Figure 4.** Diagram showing a preventive technique after tooth extraction. (**A**) Dental implantation. (**B**) Dental extraction. (**C**) Filling the dental socket with biomaterial. (**D**) Closure with a membrane. (**E**) Suturing the grafted area.

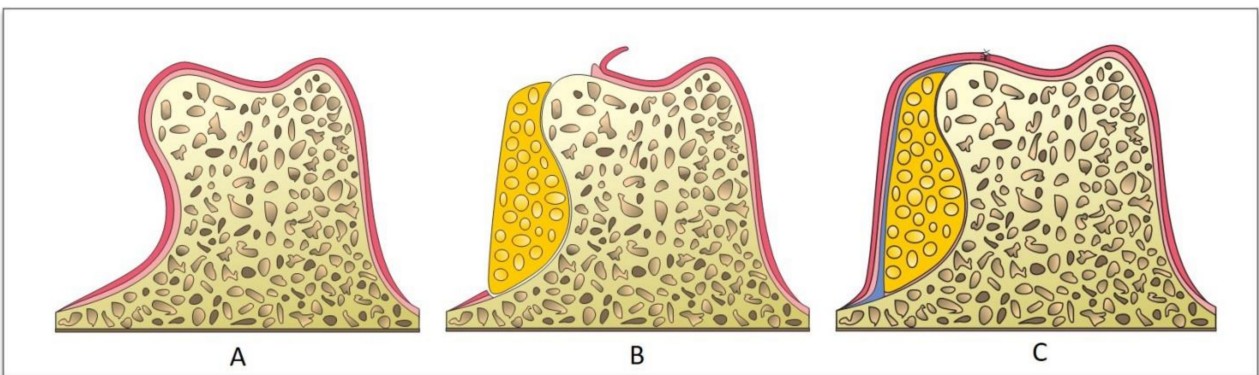

**Figure 5.** Diagram showing a regenerative technique in an area with the bone defect. (**A**) Alveolar ridge with a bone defect in thickness. (**B**) Bone graft adapted to the area of the defect and membrane positioned. (**C**) Suturing the grafted area.

Since bone repair depends on mechanisms of osteoconduction, osteoinduction, and osteogenesis, the ideal bone graft should guide the bone growth three-dimensionally, establishing cell recruitment, inducing differentiation of resident bone cells, and providing cells at the implantation site [18] (Figure 6). For many years, autogenous bone grafts had been considered the reference standard for the treatment of bone defects. In addition to having imunocompatible cells, they are osteogenic, osteoconductive, and osteoinductive presenting characteristics of bioabsorption and angiogenesis, which guarantees high clinical predictability [19,20]. While osteoinductors are biomaterials that stimulate undifferentiated cells to differentiate into osteoblasts, osteoconductors act as a framework for the proliferation of blood vessels, perivascular tissue, and osteoprogenitor cells of the patient. Osteogenitors biomaterials are capable of forming bone tissue by themselves since they have viable precursor cells and/or osteoblasts [21].

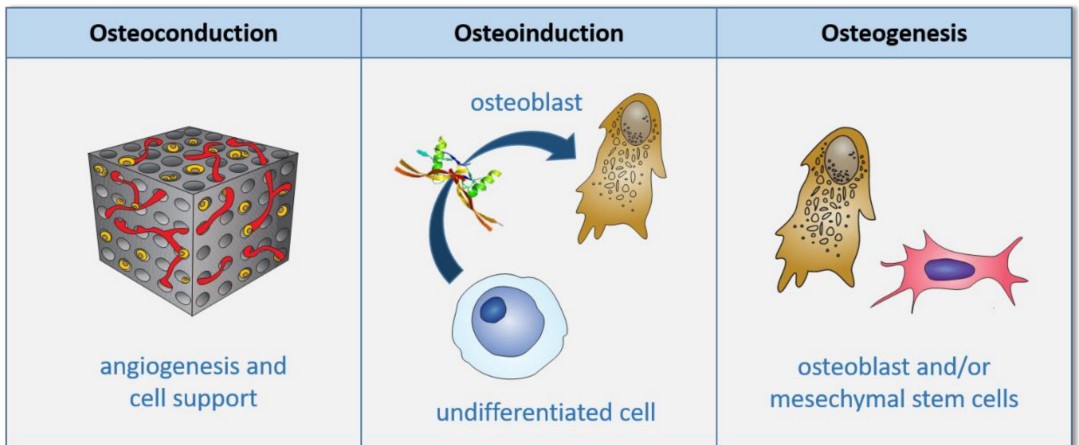

**Figure 6.** Diagram showing mechanisms of osteoconduction, osteoinduction, and osteogenesis.

In autogenous grafts, the patient is both the donor and the graft receptor. When used to correct intra-oral bone defects, autogenous bone grafts have been obtained from the oral cavity (such as the mental area, mandible body, or maxillary tuberosity) or extra-oral donor areas (such as the iliac crest, tibia, or skullcap) [19]. However, limitations such as the restricted availability of bone for removal, increased surgical morbidity, high rates of graft bone remodeling, and difficulty in predicting the rate of degradation over time, have been associated with autogenous bone grafts [22–24]. Therefore, autogenous bone grafts have declined in use over time, especially from the extra-oral area.

In order to minimize the inherent limitations of autogenous bone grafts, bone substitutes such as homologous, xenogenous, and alloplastic grafts have been suggested in the literature [19,25–27]. Homologous bone grafts are originated from another individual of the

same species from cadavers (e.g., DFDBA: demineralized freeze-dried bone allograft; FDBA: freeze-dried bone allograft), while xenogenous grafts are obtained from another species (ex: CB-BB: chemically deproteinized bovine bone; TD-BB: thermally deproteinized bovine bone). Despite being available in large amounts, the main disadvantages of homologous bone grafts are high costs, the requirement for complex sterilization and storage techniques, difficulty in predicting the rate of degradation over time, the risks of disease transmission, variable osteoinductive and osteoconductive properties, and lower osteogenic potential compared to autografts [28,29]; while risks of zoonotic diseases transmission (e.g., bovine spongiform encephalopathy), prion infections, and immunological activation of diseases have been related to xenogenous grafts [30,31].

Conversely, alloplastic bone grafts are fabricated from inorganic or synthetic biomaterials, which, despite not having the osteogenesis and osteoinduction capacity, considerably reduce surgical morbidity rates and the risk of disease transmission [32,33]. Therefore, alloplastic bone grafts have been developed to overcome the limitations of autogenous, homologous, and xenogenous grafts. Indeed, these biomaterials have demonstrated advantages such as a reduction in surgical time, an abundance of materials with no amount limitation, ease of handling, no risk of disease transmission, and very low antigenicity potential [34–36].

Alloplastic bone grafts are fabricated from absorbable or non-absorbable synthetic materials, with different sizes and shapes, and variations in their physical and chemical properties, such as granule morphology, crystalline or amorphous phase, and pore size and interconnectivity [37]. Composed of osteoconductive biomaterials, these grafts provide a framework, which will be populated by cells originated from osteoprogenitor cells (from the defect margins), thus promoting bone neoformation until the biomaterial is completely replaced by the new bone [33,35,36]. Alloplastic grafts can be used alone, or in association with autogenous bone, biomaterials, or bioactive substances [35,37,38].

## 3. Scaffolds

With the advancement and diversity of alloplastic bone grafts, the concept of "bone tissue engineering (BTE)" stands out, which aims to combine biological knowledge concerning the histology and morphology of bone tissues with the development of appropriate biomaterials and techniques for the three-dimensional (3D) structure's construction, capable of simulating the bone environment on a micro and nanoscale [39]. These structures, better known as scaffolds, are carriers for cellular interactions (migration, adhesion, and cell proliferation), allowing the deposition of a new bone extracellular matrix on its porous surface [40–42]. Additionally, they also provide temporary support for newly formed bone tissue and vessels [43,44].

Metals (tantalum, magnesium, titanium and titanium alloys, nickel-titanium alloy [nitinol]); natural polymers (collagen, gelatin, silk fibroin, chitosan, alginate, hyaluronic acid); synthetic polymers (polylactic acid [PLA], polyglycolic acid [PGA], polylactic-co-glycolic acid [PLGA], polycaprolactone [PCL], polyvinyl alcohol [PVA], polypropylene fumarate [PPF], polyurethane [PU]); bioactive ceramics (hydroxyapatite [HAp], tricalcium phosphate [TCP], beta-tricalcium phosphate [β-TCP], calcium sulfate [$CaSO_4$], akermanite [Ca, Si, Mg], diopside [$MgCaSi_2O_6$], bioactive glass [BGs]); and bioinert ceramics (aluminum oxide, zirconia) have been proposed for the manufacture of scaffolds-based bone tissue engineering [41,45,46]. Additionally, the materials most used in clinical practices to repair intra-oral defects are HAp, dicalcium, tricalcium phosphates, and bioactive glasses [47–50]. The materials used in scaffolds for implant dentistry are shown in Table 1.

**Table 1.** Materials, advantages, disadvantages of scaffolds for implant dentistry.

| Materials | | Advantages | Disadvantages |
|---|---|---|---|
| Poly-lactic acid (PLA) [46,51] | | - Water-soluble<br>- Crystallinity tunable by changing hydroxylation degree | - Non-hydrophobic<br>- Shortage of cell adhesion |
| Poly-glycolic acid (PGA) [46] | | - Water-soluble<br>- Crystallinity tunable by changing hydroxylation degree | - Non-hydrophobic<br>- Shortage of cell adhesion |
| Polylactic-co-glycolic acid (PLGA) [52,53] | | - Water-soluble<br>- Crystallinity tunable by changing hydroxylation degree<br>- Easily synthesized<br>- Biodegradable in non-toxic by-products<br>- Controlled degradation time | - Non-hydrophobic<br>- Shortage of cell adhesion |
| Polycaprolactone (PCL) [54] | | - Crosslink in situ and print by injection<br>- Elastic behavior | - Degradation rate in years |
| Polyvynil alcohol (PVA) [55,56] | | - Ability to manufacture scaffolds with various characteristics such as shape, porosity, and degradation rate<br>- Flexibility<br>- Mechanically strong<br>- Water-soluble<br>- Compatible with several polymers | - Non-soluble in organic solvents<br>- Cross-linking of polymers to maintain integrity |
| Polypropylene fumarate (PPF) [57] | | - Adjustable mechanically strong<br>- Adjustable rates of degradation | - Cross-linking of polymers to maintain integrity |

**Table 1.** *Cont.*

| Materials | | Advantages | Disadvantages |
|---|---|---|---|
| Hydroxyapatite (HAp) [44,46,58] |  | - Highly biocompatibility<br>- Nontoxic<br>- Hydrophilicity<br>- Provides calcium and phosphorus for new tissue | - Poor mechanical strength<br>- Lack of organic phase |
| Tricalcium phosphates (TCP) [59,60] |  | - Provides calcium and phosphorus for new tissue | - Poor mechanical strength<br>- Lack of organic phase |
| Bioactive glass (BG) [46,61] |  | - Bioactive<br>- Bond-bonding affinity | - High solubility<br>- Limitation of shaping |
| Zirconia (ZrO$_2$) [62,63] |  | - Mechanically strong<br>- High fracture toughness<br>- Osseointegration potential<br>- Radiopacity | - No biological activity |

More recently, hybrid scaffolds, which combine polymers and ceramics have been proposed to associate the advantages of polymeric materials with the favorable properties of ceramic such as bioactivity and mechanical resistance [41]. It is relevant to mention that there is still no single synthetic biomaterial that offers all the desirable properties for a scaffold; thus, the association of biomaterials combines the best properties of each one, in order to meet the needs of the bone-grafted area [48]. These 3D structures can also be associated with growth factors (such as recombinant human bone morphogenetic protein-2 (rhBMP-2) and platelet-derived factors-BB), bioactive substances (such as simvastatin), and specific cells which stimulate bone tissue regeneration (such as mesenchymal stem cells or osteoblasts) [64–66].

## 4. Expected Properties for a Scaffold

Ideally, a scaffold should be composed of biocompatible materials that are easily adaptable to the bone defect; must present controlled bioabsorption and in line with bone formation; promote high surface wettability inducing cell adsorption and proliferation; having appropriate surface chemical properties to enable cell adhesion; have an interconnected pore architecture; present satisfactory mechanical properties to support intra-oral loads in the defect area; allow sterilization, and present industrial viability to be manufactured in different sizes and shapes [65–69].

Therefore, during the manufacture of a scaffold, the biomaterial design must be taken into account, the desired morphology, pore size and interconnectivity, and the mechanical properties [49,69]. If the scaffold conformation is similar to the defect shape, its adaptation will be more effective, quickly establishing a solid interface and the complete integration between the surface of the biomaterial and the bone tissue [70]. This intimate adaptation, associated with the chemical properties of the surface and a porous structure interconnected by the association of macro and micropores, facilitates cell dynamics, enabling the adhesion, proliferation, and migration of bone-related cells with subsequent deposition of osteoid tissue inside the scaffold [46,70,71].

However, although the importance of the presence of porosities to create a microenvironment for cell proliferation is clear, there is still no consensus on the ideal pore size for a scaffold in bone repair. The literature presents different values, ranging from 40 to 500 μm in diameter [41,71,72]. These variations are probably related to the nature and variability of the bone (cortical and/or spongy) where the bone graft will be used and the biomaterial used to manufacture the scaffold [72,73].

Studies suggest that smaller pores (with around 40 μm) favor cell agglomeration, while larger pores (with approximately 100 μm) accelerate cell migration [74,75]. It is recommended to use pores with at least 100 μm in diameter to ensure successful diffusion of nutrients and oxygen, which enables cell survival and stimulates bone growth [76,77]. Other studies indicate that diameters of 5 μm are suitable for neovascularization, 5–15 μm for fibroblast growth, 20 μm for hepatocyte growth, 20–125 μm for skin regeneration in adult mammals, 40–100 μm for osteoid tissue growth, and diameters between 100–350 μm favor bone regeneration [78]. Additionally, the use of nanoscale pores can increase the surface area, which is advantageous for the apatite formation and proteins and/or osteoblasts fixation [79,80]. Interconnectivity among the pores must also be obtained, since a network of interconnected pores increases the diffusion rates to the center of the scaffold, allowing vascularization, improving the transport of nutrients and oxygen, and facilitating the removal of metabolic waste [76,77].

It is also reported that porous scaffolds with nano, micro, and macroporosities, can perform better than macroporous scaffolds [81]. However, the reproduction of varying degrees of porosity increases the complexity and the challenge in making reproducible scaffolds, especially when composed of a single material [68]. Additionally, high porosity scaffolds demonstrated a reduction in the mechanical properties, directly impacting its structural integrity [68,82]. The mechanical characteristics of a scaffold must be similar to

those of native tissue, especially concerning the resistance to stresses suffered in vivo, until the new tissue formed occupies the scaffold matrix [82].

The scaffold degradation capacity and speed are also important parameters to consider. Ideally, the scaffold degradation should be concomitant with the bone formation process, since its mechanical properties decrease as a result of its degradation [83]. In physiological and artificial aqueous environments, biomaterials can degrade via some mechanisms, including (i) physicochemical degradation (chain scission and dissolution in an aqueous environment), (ii) enzymatic activity, (iii) cellular degradation (e.g., inflammation, foreign body response), and (iv) mechanical fragmentation due to a loss of structural integrity resulting from the former mechanisms [84,85]. The rate of scaffold degradation is determined by factors such as the configurational structure, crystallinity, molecular weight, morphology, stresses, porosity, and implantation site [86]. Polymeric and resorbable ceramic biomaterials can be degraded; however, polymeric scaffolds have higher degradation rates [87].

## 5. Polymeric Biomaterials Applied to Fabricate Scaffolds Used in Implant Dentistry

Polymeric materials are composed of a long repeating chain of monomers formed by covalent bonds. Natural polymers such as proteins (fibroin, collagen, gelatin, fibrinogen, elastin, keratin, actin, myosin), polysaccharides (cellulose, amylose, dextran, chitosan, glycosaminoglycans), and polynucleotides (DNA, RNA) have a better interaction with biological systems due to its bioactive, bioadhesive, and hydrophilic properties [87], while synthetic polymers (synthetic aliphatic polyesters: poly-glycolic acid (PGA), poly-lactic acid (PLA), poly-lactic acid-co-glycolic acid (PLGA), polycaprolactone (PCL) demonstrated more possibilities for chemical and mechanical modifications [51].

The polymer's biodegradation occurs by hydrolytic scission in its main chain, resulting in soluble, non-toxic oligomers or monomers. The main biodegradation processes occur by two mechanisms: (i) hydrolysis or enzymatic digestion of the main chain, promoting gradual erosion of the polymer, and (ii) rupture of the crosslinking links, generating water-soluble fragments, which are transported away from the site deployment [84,85].

### 5.1. Poly-Lactic Acid (PLA)

PLA is a polyester obtained by condensing hydroxyl and carboxyl groups of the lactic acid monomer or by opening the lactide ring. In addition to its biocompatibility and biodegradability, it has low rigidity, good processability, and is thermally stable [88]. Chemically, PLA is considered an organic acid with an asymmetric carbon that has two enantiomers (L+ and D−) and a racemic DL [51], being used as a precursor in the manufacture of polymers. Approved for use in the biomedical area since the 1970s due to its biocompatibility, when PLA comes into contact with the human body, it undergoes hydrolytic degradation via a mass erosion mechanism by a random splitting of the ester bonds, decomposing into lactic acid and producing water and carbon dioxide via the Krebs cycle. Its degradation depends on characteristics such as degree of crystallinity, molar mass, type of isomerism, and changes in pH [89]. In the treatment of peri-implant, periodontal, and bone defects, PLA is used in the form of membranes.

### 5.2. Poly-Glycolic Acid (PGA)

PGA is a biodegradable, thermoplastic polymer, and the simplest linear aliphatic polyester. It can be prepared to start from glycolic acid using polycondensation or ring-opening polymerization. However, high-molecular-weight PGA could not be obtained because it was unstable and easily degradable compared with other synthetic polymers [90]. Polyglycolide fiber is a clinically well-known non-woven fabric, which has rapid absorption as an advantage [91]. Conversely, the polyglycolide mesh has low integrity mechanics in vitro. Therefore, its application in bone in an isolated form is inadequate. PGA combining with materials that promote a greater reinforcement to bone tissue can obtain a stable combination. The association of the PGA mesh with a PLLA solution allowed a

substantial increase in resistance to compression than PGA alone [92]. Currently, PGA and its copolymers poly(lactic-co-glycolic acid) with lactic acid, poly(glycolide-co-caprolactone) with ε-caprolactone, and poly (glycolide-co-trimethylene carbonate) with trimethylene carbonate have been widely used as biomaterials in the biomedical field [93].

### 5.3. Polylactic-Co-Glycolic Acid (PLGA)

PLGA is a biocompatible biomaterial, easily synthesized, and biodegradable in non-toxic by-products [52,53,94]. This copolymer has been used in a variety of therapeutic devices approved by the Food and Drug Administration due to its high rate of biodegrad-ability and biocompatibility. PLGA is synthesized by the ring-opening copolymerization of two different monomers, PGA and PLA. Therefore, modifications of physical-chemical characteristics can be performed by the composition of the original monomers (PLA and PGA). Depending on the ratio of PLA to PGA used in the polymerization, different forms of PLGA can be obtained. These forms are usually identified concerning the molar proportion of the monomers used (e.g., PLGA 75:25 identifies a copolymer whose composition is 75% PLA and 25% PGA). The crystallinity of PLGAs will vary from 100% amorphous to 100% crystalline depending on the block structure and molar ratio [95].

Since PLGA is highly biocompatible and non-toxic, in addition to being easily processed in different devices, the clinical applications of PLGA have increased in recent years, especially in the field of orthopedics as devices for fixing fractures in the craniomaxillofacial region, support for cell growth, and a device for controlled drug release [96]. Additionally, the PLGA membrane is also indicated for periodontal, peri-implant, and bone regeneration. It is important to mention that although PLGA is not considered osteoinductive, it allows the incorporation and release of biomolecules with substantivity [96].

PLGA is degraded faster than PLA because of glycolic acid incorporation in the polymer chain through de-esterification. PLGA scaffolds are often used as bone reconstruction materials. They can be synthesized in personalized shape and to satisfy the required absorption time. There are various methods for processing these porous synthetic scaffolds. Nevertheless, PLGA has demonstrated reduced cell adhesion and proliferation in response to its hydrophobicity [54,97].

### 5.4. Polycaprolactone (PCL)

PCL consists of hexanoate units and represents an important biodegradable aliphatic polymer. It is synthesized by poly-condensation of 6-hydroxyhexanoic acid and ring-opening polymerization of ε-caprolactone [54]. Due to the interconnected pores, high porosity, and elastic behavior, the 3D DPCL electrospun nanofibrous has a similar structure to the extracellular matrix and has demonstrated unique features for tissue formation [98,99]. PCL has been used to fabricate several types of hybrid scaffold [93,99].

### 5.5. Polyvinyl Alcohol (PVA)

PVA is a biodegradable synthetic polymer which is synthesized by hydrolysis of poly(vinyl acetate [55,56]. Some unique features of PVA (e.g., solubility, flexibility, biocompatibility, biodegradability, mechanical strength) make it an important choice as a polymeric scaffold for bone tissue engineering.

This polymer is interesting for electrospinning due to the presence of a hydroxyl group in its repeating unit, which makes it cross-linkable using its interconnected hydrogen bonding [100,101]. PVA is the most commonly used water-soluble synthetic polymer for biomedical applications [100]. PVA is not soluble in organic solvents and only sparsely soluble in ethanol. Due to PVA compatibility with several polymers, it can be easily mixed up with several biomaterials, extending its applicability. Different studies demonstrated that the mechanical property of PVA can be improved without compromising the degradability through the inclusion of reinforced agents [102,103].

The physicochemical property of PVA is determined by the degree of hydrolysis during the synthesis procedure. Because PVA is a water-soluble polymer, before any biological

application, cross-linking of polymers is important to maintain integrity. Therefore, the degree of cross-linking plays an important role in deciding the stability in the biological environment, fluid uptake, degradation property, among others. For biomedical applications, physical cross-linking is more useful as it does not leave any residual toxic crosslinking agents [56,100].

### 5.6. Polypropylene Fumarate (PPF)

Since its introduction by Yaszemski et al. [57], PPF has been used preclinically for bone regeneration. PPF demonstrates several medical requirements including biocompatibility, mechanical properties, osteoconductivity, and capacity to be sterilized [57,104,105]. This synthetic polymer degrades via hydrolysis of its ester bonds. Additionally, the degradation time depends on the molecular mass of the backbone chain, the types of crosslinker used, and the crosslinking density [104,105]. PPF is degraded in non-toxic fumaric acid and propylene glycol, equal favorable for in vivo applications [106]. In PPF cross-linked, the strength is adequate to guide and allow cell attachment and tissue formation in vivo. Moreover, the PPF degradation occurs in a timeframe adequate to bone healing and remodeling [107].

### 6. Ceramic Biomaterials Used in Scaffolds Applied in Implant Dentistry

Ceramics are inorganic, non-metallic, and crystalline materials, which can be classified as bioinert and bioactive. Bioinert ceramics have no interaction with living tissue, while bioactive ceramics are capable of promoting adherence to living bone tissue [108]. The ceramics most used in bone tissue engineering are bioactive, also known as bioceramics, with emphasis on hydroxyapatite and $\beta$-tricalcium phosphate [109,110].

These bioceramics contain calcium salts that stimulate the formation and precipitation of calcium phosphates in bone tissue [111]. However, due to their low structural rigidity, they cannot be used in areas of great mechanical stress, because of the risk of fracture [112]. To address these mechanical limitations, bioinert ceramics, such as zirconia, have been suggested for use alone or associated with bioactive ceramics [113].

### 6.1. Hydroxyapatite (HAp)

Hydroxyapatite, a hydrated calcium phosphate ($Ca_{10}(PO_4)_6(OH)_2$), is a mineral present in vertebrates (about 55% of the bone composition, 96% of dental enamel composition, and 70% of dentin), which acts as a reserve of calcium and phosphorus [58,110]. For use as a graft material, it can be obtained by deproteinizing bone tissue (natural HAp, usually from bovine tissue) or by precipitating aqueous solutions from phosphates (synthetic HAp) [114]. Natural and synthetic HAp are thermodynamically stable at physiological pH and actively participate in bone bonds, forming a strong chemical bond with bone tissue [58]. The HAp surface allows the interaction of dipole-type bonds, causing water molecules and also proteins and collagen to be adsorbed on the surface, thus inducing tissue regeneration [59].

Synthetic HAp has been the most widely used clinically, characterized by being a biocompatible and osteoconductive material that presents high stability in aqueous media [115]. It is commercialized in the form of dense or porous ceramics, in blocks, granules, or coatings, being used in the repair of bone defects, an increase of alveolar ridge, guided regeneration of bone tissues, and buccomaxillofacial reconstructions [116,117]. Compared with natural HAp, synthetic HAp has a higher crystallinity, which results in slower degradation that can last 4 to 5 years [116]. Therefore, scaffolds manufactured in HAp maintain their geometric shape for a longer time during the regeneration of bone tissue [117].

However, in some clinical situations, the rate of HAp degradation may be out of step with bone formation [118]. When compared to other calcium phosphates (amorphous tricalcium phosphate: 25.7 to 32.7 g/L; calcium monophosphate monohydrate: about 18 g/L; anhydrous calcium monophosphate: about 17 g/L), the rate of HAp reabsorption

is considered to be quite low (about 0.0094 g/L) [119,120]. Thus, studies have suggested replacing phosphate groups ($PO_4^{3-}$) with carbonate groups ($CO_3^{2-}$) (carbonated or carboapatite HAp), which modifies the crystalline structure of HAp, increasing its solubility and, consequently, its clinical application [116,120].

### 6.2. Tricalcium Phosphates (TCP)

When subjected to high-temperature treatments, HAp can give rise to other phases such as tricalcium phosphates (α and β-TCP) that are also frequently used as bioceramic materials. α-TCP and β-TCP have the same chemical composition ($Ca_3(PO_4)_2$); however, the crystallographic structures are different, and the α phase is more soluble. Additionally, α- and β-TCP have different densities: α-TCP (2.86 g/cm$^3$) and β-TCP (3.07 g/cm$^3$); the last being closer to that of HAp (3.16 g/cm$^3$) [59].

Biomaterials composed of calcium phosphate (CaP) can be manufactured in both porous and dense forms as bulk, granules, and powders, besides the de-coating form. These biomaterials demonstrated biocompatibility, safety, availability, low morbidity, and are affordable. CaP bioceramics are now in common use for different medical and dental applications such as treatment of bone defects and fractures, total joint replacement, spinal surgery, dental implants, peri-implants and periodontal therapy, and craniomaxillofacial reconstruction [121].

CaP-based biomaterials are bioactive and have a composition and structure similar to the mineral phase of bone. Despite the osteoconductive property [60], CaP-based biomaterials have a high affinity for protein adsorption and growth factors [122]. The osteoinductive property can be achieved by: (i) structural or chemical optimization of the biomaterials themselves; and/or (ii) incorporation of osteoinductive substances, such as rhBMP [123,124].

Notwithstanding the several advantages of CaP bioceramics, these biomaterials demonstrated poor mechanical strength, lack of organic phase, presence of impurities, micro-scale grain size, non-homogenous particle size and shape, prolonged fabrication time, and difficult porosity control [125]. However, several modifications of fabrication parameters have been performed and the physicochemical properties of these biomaterials are thereby improved [126].

### 6.3. Bioactive Glass (BG)

Bioactive glass (BG) was first developed by Hench et al. in 1971, with the $4_5S_5$ composition through the use of $Na_2O$-$CaO$-$SiO_2$-$P_2O_5$ phase diagram [61], which demonstrated biocompatibility and bone-bonding ability. These synthetic materials based on silica are highly bioactive, due to the calcium and phosphate ions in their composition [127]. When BGs are exposed to bone or biologic fluids, their structure fully reacts to form internal silica gel cores with calcium phosphate-rich surface. Therefore, the internal silica gel core degrades, leaving an external calcium phosphate bulk, which is structured as a hydroxy-carbonated apatite layer that improves protein adsorption to BGs' surface and integration with surrounding tissue [128,129]. The Ca:P ratio, composition, and microstructure of BGs determines the rate of ion release from the BGs' surface.

Inside the degraded BGs, osteoprogenitor cells differentiate and form new bone. BGs are particularly attractive for bone repair due to their controllable degradation, osteogenic potential, and bone-bonding affinity [130]. It is relevant to mention that BGs degradation rate is highly tunable due to changing their chemical compositions or material processing methods. Therefore, BGs can be designed with a specific degradation rate to respond to the precise requirement of a certain bone repair.

Many variations of BGs are currently being used in periodontics and implantology. They are generally composed of silica (45%), calcium oxide (24.5%), sodium oxide (24.5%), and pyrophosphate (6%), named $4_5S_5$. Clinically, this composition of BGs has been used in restorative dentistry, periodontics, implantology, and maxillofacial area for periodontal,

peri-implant, and bone defects [131,132]. Recently, mesoporous BGs have been developed, which enables greater degradation control [133].

### 6.4. Zirconia (ZrO$_2$)

Zirconia is a structural ceramic that has been used for biomedical applications due to its biocompatibility, osseointegration potential, radiopacity, favorable mechanical properties, and in particular, its toughness [134–136]. When a crack occurs in zirconia, an internal tension is generated due to its propagation, transforming some grains from tetragonal to monoclinic (t→m), which increases the volume by about 5% [137]. As a result, compressive stress is generated, acting on the crack tip and hindering its propagation [137,138]. This phenomenon of "containment" of the crack is known as "transformation toughening", and since the discovery by Garvie et al. [139], it has been the focus of research for the biomedical application of zirconia.

Due to this favorable behavior, zirconia can supply the mechanical needs of a scaffold, so that it does not deform when submitted to loading and can be used to increase atrophic alveolar ridges or to replace the bone loss in the maxillofacial area [62]. Additionally, zirconia scaffolds can be manufactured by various techniques, resulting in different degrees of porosity, control of the geometric structure, and micro-roughness, which allows a good interconnection structure between the pores to support the growth of osteoblasts, vessels, and new bone [63,140].

However, despite offering superior properties, such as corrosion resistance, low friction coefficient, great wear resistance, hardness, and resistance to fracture propagation, zirconia scaffolds do not have the same efficiency in integration with bone tissue as phosphate-based ceramics [141]. Thus, nanocrystalline calcium phosphate powders, tricalcium phosphates, and/or bioactive glass have been associated with zirconia scaffolds, in the form of coatings or infiltrations, to increase biological activity, healing capacity, and osteogenesis within the adjacent tissue [138,140,142–144]. The current trend of using hybrid scaffolds, through the association of different materials, has been the path that tissue bone engineering has been seeking to obtain artificial structures more similar to bone biology.

## 7. Techniques for Manufacturing Scaffolds

Due to the several biomedical areas that benefit from tissue bone engineering, the rapid advance in the manufacture of 3D structures has been accompanied by the development and improvement of methods that aim to achieve the desired criteria for a scaffold. Scaffolds can be manufactured by conventional or additive manufacturing techniques and more recently, by 3D and 4D printing techniques [145,146]. Conventional techniques include methods such as solvent casting and particle leaching, freeze-drying, thermally induced phase separation, gas foaming, powder-forming, polymeric sponge replica method, and electrospinning [145,147–150], while among additive manufacturing techniques stereolithography, fused-deposition modeling, selective stand out laser sintering and electron beam melting stand out [145,151].

### 7.1. Conventional Techniques

Conventional techniques for manufacturing scaffolds use subtraction methods, in which part of the materials is removed so that the desired properties are achieved [152]. Generally, these techniques are easy to made and present low cost; however, these techniques may have limitations, such as the difficulty of obtaining structures with complex geometries [73]. The chemicals in the solvents used may not be completely removed from the scaffolding, being toxic to the newly formed tissue and the surrounding tissue of the host [153]. Table 2 describes the most commonly used conventional techniques and the scaffolds that can be obtained from them.

**Table 2.** Conventional manufacturing techniques: description and typical scaffold materials.

| Technique | Description | Scaffold Materials |
|---|---|---|
| Solvent casting and particle leaching [154] | A polymer solution is dissolved in a solvent rich in crystals of soluble salts or organic particles. After removing the solvent by evaporation, these particles come together to form a matrix. The system is immersed in water, allowing the dissolution of the salt matrix and the removal of the produced polymeric structure, which is highly porous. The structures produced are simple but may contain some solvent residue. The centrifugation and layer technique can be combined to minimize these limitations. | - PLGA |
| Freeze drying [155,156] | The polymeric material is dissolved in a solvent and the solution obtained is cooled below its freezing point taking the solvent to solidification. This system is taken to a freeze dryer, previously adjusted with a temperature below the freezing point of the solvent and a pressure below atmospheric pressure to promote the sublimation of the solvent. The result is the formation of a porous structure, with multiple empty spaces and channels connected. | - Gelatine<br>- HAp<br>- PLA<br>- PCL<br>- Chitosan |
| Thermally-induced phase separation [148,157] | A polymer is dissolved in a solvent at high temperature, followed by rapid cooling. The solvent is separated from the polymeric structure due to the change in the solubility coefficient caused by the temperature reduction, forming one phase rich in polymer and another poor. The polymeric phase solidifies, while the other phase is removed, resulting in a highly porous polymeric structure. This technique can be used in association with other techniques to manufacture 3D structures with controlled pore morphology, such as leaching. | - PPLA<br>- Chitosan |
| Gas foaming [158] | Blowing agents are used to pressurize molded polymers. These agents generate gas bubbles that act as porosity builders, causing expansion in volume and reduction in the density of polymers. When associated with the replica technique, the polymeric foam is impregnated with a ceramic suspension. The structure sintered at high temperature, degrades the polymer, resulting in a porous ceramic structure. | - HAp<br>- β-TCP |
| Powder-forming [116] | A suspension of ceramic particles is prepared in an appropriate liquid to form a paste. From this paste, green bodies are produced in different ways. Subsequent sintering results in porous scaffolds. | - PLGA<br>- HAp |
| Electrospinning [159,160] | An electric field is used to form fibers with diameters ranging from micrometer to nanometer scale. A typical apparatus consists of an infusion pump, syringe set, and metallic needle for the formation of the spinning droplet, a collector, and the electrical system. The potential difference applied by the electrical system generates high electric fields and its strength exceeds the surface tension of the droplet, elongating it. After evaporation of the solvent, the fibers are collected. | - PLA<br>- β-TCP |

### 7.2. Additive Manufacturing Techniques

Additive manufacturing (AM), commonly known as 3D printing (3DP), includes techniques based on the traditional principles of rapid prototyping, which are used to manufacture a physical object, using three-dimensional computer-aided data (CAD) [161]. They employ additive processes, where the manufacture of three-dimensional physical models is undertaken layer by layer. The production of parts with low volume and with the high complexity of format is facilitated, due to better control of properties such as compressive strength, elastic modulus, dissolution, and mass transport [160,162]. Also, a specific geometry, with a particular shape, size, and porosity (uniform or functionally graded) can be achieved.

AM techniques have the advantage of manufacturing patient-specific designs, which can be obtained from the computed tomography scan of the bone defect. This is particularly important when repairing more complex injuries [163]. These techniques do not use toxic organic solvents and allow better control of pore architecture, pore volume, and percentage porosity, in addition to the mechanical properties of the scaffold. Thus, AM techniques are superior to conventional methods, where it is difficult to control the pore size, shape, and pore interconnectivity [152]. Moreover, AM techniques increasingly allow the manufacture of hybrid scaffolds, combining the advantages of the selected materials [30,48,113,142].

AM or 3DP techniques—such as stereolithography (SL), fused-deposition modeling (FDM), and selective laser sintering (SLS)—combine computer-aided design (CAD), computer-aided manufacturing (CAM), and computer numerical control (CNC) [164]. More recently, additively manufactured structures using smart (intelligent) materials that can modify in a pre-defined form or perform a pre-defined function according to the stimuli are characterized in "4D printing" processes [146]. The techniques used are the same as those mentioned for 3DP; however, the nature of the materials used are different, which

must present "shape memory" or "self-performance" [165]. Table 3 describes the most widely used AM techniques and which scaffold materials can be obtained from them.

**Table 3.** Additive manufacturing techniques: description and typical scaffold materials.

| Techniques | Description | Scaffold Materials |
|---|---|---|
| Stereolithography (SL) [166,167] | Solid objects are manufactured, layer by layer, by curing a photoreticulable liquid resin of ultraviolet or visible light beams, directed by a dynamic mirror system. A mobile platform moves the cured part. Therefore, another layer can solidify producing a three-dimensional structure. | - PEG - PPF |
| Fused Deposition Modeling (FDM) [168] | Thermoplastic filaments, consisting of an extruded material or composite, are melted and deposited layerwise on a build platform until the object is formed. | - ABS - PLA - nylon |
| Selective Laser Sintering (SLS) [169,170] | In this technique, also known as selective laser melting (SLM), the poorly compacted powder is sintered with a high-power laser (e.g., $CO_2$), particle by particle, uniting them in a controlled manner, forming thin layers. The layers are joined to each other according to predefined computer-aided data (CAD) parameters. The interaction of the laser beam with the powder increases the temperature of the powder above the glass transition temperature and below the melting temperature, causing the melting and bonding of the particles to form a solid mass. The process results in solid or porous structures with superior mechanical properties, custom density, and elastic modulus, and a post-processing phase is required to remove the remaining power. | - PPLA - HPa |
| Bioprinting [171] | Cells and biomaterials are printed using inkjet, extrusion, or laser-assisted bioprinting techniques with micrometric precision. Jet-based bioprinting produces 2D and 3D structures by applying layers of bio-ink on a substrate. In extrusion-based bioprinting, a mixture of hydrogels is injected on pressure. Afterward, the hydrogels are solidified physically or chemically, and the 3D structures are manufactured by stacking. In laser bioprinting, a receptor material made of glass covered with a layer of gold absorbs the laser, and in this way, a drop is created at high pressure, which in turn transfers materials to the substrate. | - alginate - chitosan - collagen - fibrin |

## 8. Future Studies

The progress of scaffolds for bone formation during the last few decades has been remarkable. As described in this review, scaffolds can be composed with different materials and combinations, as well as, using several manufacturing techniques. Due to the notable developments in biotechnology and manufacturing technologies in the last few years, emergent smart scaffolds have been arising. However, the clinical application of some of such scaffolds needs time. It is necessary to further clarify the interaction between the surface of the scaffolds and tissues and study the degradation process of such materials in different kinds of human bone (trabecular/cortical, different densities in different age groups). Moreover, it is arduous to understand all these biological events in depth, especially taking into account that in some situations the scaffolds will be grafted simultaneously to the dental implant or that, after the grafting procedure, the dental implant will be installed.

## 9. Conclusions

In summary, conventional and 3D printing manufacturing techniques and associated materials are revolutionizing the development of biomaterials for scaffolds in implant dentistry. Clinical applications include patient-specific implants and prostheses; engineering scaffolding for the regeneration of tissues, and customization of drug-delivering systems. Currently, there are only a limited number of biodegradable materials available for the manufacture of materials and composites, particularly by 3D printing techniques. Therefore, there is a great need for research to manufacture new biomaterials and biocomposites with adjustable properties that can restore functionality at the application site. Low-cost and readily available lactic acid-based polymers (such as PLA and PCL) are focused, mainly due to their ability to work well in most types of 3DP technologies. Also, they have excellent mechanical and biodegradable properties. These polymers can be mixed with ceramic biomaterials (such as HAp, TCP, bioglass) and used as composites to provide greater printability, mechanical stability, and better integration of tissues for dentistry applications.

**Author Contributions:** M.Ö.: Critically revised and edited the manuscript for important intellectual content. D.H.: Edited and revised the manuscript for important intellectual content. M.C.F.: Drafted the manuscript and revised the manuscript for important intellectual content. A.C.: Contributed to the acquisition and analysis of articles; drafted the manuscript. C.A.M.V.: Contributed to the acquisition and analysis of articles; drafted the manuscript. All authors have read and agreed to the published version of the manuscript.

**Funding:** This research received no external funding.

**Conflicts of Interest:** The authors declare no conflict of interest.

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
