# Peer review of "Materials and Manufacturing Techniques for Polymeric and Ceramic Scaffolds Used in Implant Dentistry"

_jcs, doi:10.3390/jcs5030078_

Round 1

Reviewer 1 Report

Evaluation of the “Materials and Manufacturing Techniques for Polymeric and Ceramic Scaffolds used in Implant Dentistry”

1Overall this work is interesting and good presentation style

Some comments to further improve it :

2In introduction it is worth stressing the need of this review and how this is different from other work; also at the end of introduction it will be worth indicating how this work is structured for easy following the paper

3I suggest considering also the manufacturing of “bioactive glass scaffolds”: https://doi.org/10.1016/j.jmbbm.2020.103854  ; in line 151

4I suggest endorsing the data from Table 1 with some citation which will make this table more robust, cause now will be difficult for readers to search some sources for these materials

5For Techniques for manufacturing scaffolds you can consider also “https://doi.org/10.1016/j.matdes.2020.109026” that was presented very recently in.

6Please put a reference for each method in Table 2

7Please put a reference for each method in Table 3 cause the researchers knows about them but not sure if they were applied in this field

8Few lines with future work will make this review more attractive

9Also I suggest to cite more recent references as now from 100+ only few are very recent one

Author Response

- In introduction it is worth stressing the need of this review and how this is different from other work; also at the end of introduction it will be worth indicating how this work is structured for easy following the paper.

*The need for this review and its structure were included at the end of the introduction.

- I suggest considering also the manufacturing of “bioactive glass scaffolds”: https://doi.org/10.1016/j.jmbbm.2020.103854

*This study is included in the present review.

- I suggest endorsing the data from Table 1 with some citation which will make this table more robust, cause now will be difficult for readers to search some sources for these materials

*Adequate references were included in Table 1.

- For Techniques for manufacturing scaffolds you can consider also “https://doi.org/10.1016/j.matdes.2020.109026” that was presented very recently in.

*This study is included in the present review.

- Please put a reference for each method in Table 2.

*Adequate references are included in Table 2.

- Please put a reference for each method in Table 3 cause the researchers knows about them but not sure if they were applied in this field.

*Adequate references are included in Table 3.

- Few lines with future work will make this review more attractive.

*Few lines with future work are included at the end of the review.

- Also I suggest to cite more recent references as now from 100+ only few are very recent one.

*More recent references are included in this review.

Reviewer 2 Report

The paper is interesting, written in a clear and concise way.  It is a valuable contribution in the field  

Author Response

We appreciate the considerations made by this reviewer to our manuscript.

Round 2

Reviewer 1 Report

Thank you